# MetaTeacher: Coordinating Multi-Model Domain Adaptation for Medical Image Classification

**Zhenbin Wang**[1], **Mao Ye**[1]*, **Xiatian Zhu**[2], **Liuhan Peng**[3], **Liang Tian**[1], **Yingying Zhu**[4]

[1]University of Electronic Science and Technology of China, Chengdu, China
[2]University of Surrey, Guildford, UK
[3]Xinjiang University, Ürümqi, China
[4]University of Texas, Arlington, US

*zhenbinwang@foxmail.com, cvlab.uestc@gmail.com, xiatian.zhu@surrey.ac.uk*

## Abstract

In medical image analysis, often we need to build an image recognition system for a target scenario with the access to small labeled data and abundant unlabeled data, as well as multiple related models pretrained on different source scenarios. This presents the combined challenges of multi-source-free domain adaptation and semi-supervised learning *simultaneously*. However, both problems are typically studied independently in the literature, and how to effectively combine existing methods is non-trivial in design. In this work, we introduce a novel MetaTeacher framework with three key components: (1) A learnable coordinating scheme for adaptive domain adaptation of individual source models, (2) A mutual feedback mechanism between the target model and source models for more coherent learning, and (3) A semi-supervised bilevel optimization algorithm for consistently organizing the adaption of source models and the learning of target model. It aims to leverage the knowledge of source models adaptively whilst maximize their complementary benefits collectively to counter the challenge of limited supervision. Extensive experiments on five chest x-ray image datasets show that our method outperforms clearly all the state-of-the-art alternatives. The code is available at `https://github.com/wongzbb/metateacher`.

## 1 Introduction

Despite the great stride made by existing deep learning methods on medical image classification results [32, 53, 67], their performances often degrade drastically when applied to a new unseen scenario. This is mainly due to the domain shift challenge between the training and test data, caused by different environments, different instruments, and different acquisition protocols. Unlike natural images, annotating medical images requires special clinical expertise. It is hence more difficult to obtain large-scale medical image datasets with high-quality labels at every single scenario. Domain adaptation is a feasible solution, but comes with several limitations. Firstly, medical data is often under strict privacy and license constraints. That means the source domain data is usually inaccessible during domain adaptation. Secondly, medical data is typically multi-labeled which means that there are multiple labels for a sample, and the multiple categories are not mutually exclusive. It has more prominent different characteristics in different scenarios. Considering these practical constraints, we propose a new *Semi-supervised Multi-source-free Domain Adaptation* (SMDA) problem setting in the context of medical image classification. Our proposed setting has three key conditions: (1) There are multiple source domain models trained on respective multi-label medical image datasets; (2) All

---

*The corresponding author.

the source domain data is inaccessible for adaptation; and (3) The target domain data has only a small number of labelled samples along with abundant unlabeled data.

In medical image classification, there are limited domain adaptation works, with a need of accessing the source domain data [5, 19, 24, 34, 45, 55, 57, 62]. Further, they usually consider a single source domain. On the other hand, for employing multiple source domains, existing Multi-Source Domain Adaptation (MSDA) methods typically learn a common feature space for all source and target domains [58] or use ensemble methods combined with source classifiers [8]. However, all of these MSDA methods require access to the source domain data. Regarding multi-label medical image classification, there exists a solution which extends the standard classifier network by conditional adversarial discriminator networks [46]. But it is still not source-free. Indeed, there have been extensive study on Source-Free Domain Adaptation (SFDA) [35, 64]. However, they are not directly applicable to our problem. Firstly, most of them assume a single source domain [35, 64]. Using a SFDA method to transfer each source domain model to the target domain separately and average their predictions, this strategy cannot reveal the complementary information between different source domains. Secondly, the source model is often domain biased. Different hospitals are featured with different populations, leading to a situation that the source datasets focus on a specific set of class labels. The existing SFDA methods can not assess the credibility of a source domain model with different labels.

To address the above SMDA's limitations, employing knowledge distillation from multi-source models to the target domain can be considered [18, 42, 65, 69, 70]. This forms a multi-teacher and one-student scheme. In our problem setting, a few labels of the target domain are provided to judge the credibility of multi-source models in different labels. In reality, it is common to exploit a few labeled data in the target domain. Recent works [25, 29, 50, 51] have shown that a few labeled data from the target domain can significantly improve the performance of the model. Inspired by meta-learning approaches [40, 47, 49], we consider a bilevel optimization strategy to update both the teachers and students. This is because different models vary in reliability and there is a need for optimizing the update direction for each source model. This offers an opportunity of leveraging the complementary and collaboration of different source models during model optimization, critical for solving the low-supervision challenge.

Based on the above analysis and consideration, we propose a novel framework, namely **MetaTeacher**. Specifically, it is based on multi-teacher and one-student models. Each teacher model is pre-trained on a specific labeled source data. The student model is initialized by a randomly chosen teacher. In order to provide different update directions for multiple teachers, a coordinating weight learning method is proposed to determine the contribution of each teacher for each target sample. In addition to knowledge transfer from multiple teachers, when adapting a specific teacher model, we also explore the feedback from the student and other teachers in a semi-supervised meta learning manner [16, 47]. Unlike the previous MSDA approaches, MetaTeacher can adapt each teacher in different directions according to the learned coordinating weight. This enables us to fully use different characteristics of source models, whilst avoiding the problem of insufficient training samples for multi-label classification to some extent.

Our **contributions** are summarized as follows: (1) We propose a new problem setting, *i.e.*, semi-supervised multi-source-free domain adaptation for multi-label medical image classification. To our best knowledge, our work is the first attempt at multi-source-free and semi-supervised domain adaptation in the field of transfer learning. (2) A novel framework, MetaTeacher, based on a multi-teacher and one-student scheme is introduced to solve the proposed SMDA problem. A mutual feedback mechanism is designed based on meta-learning between the target model and the source models for more coherent learning and adaptation. The knowledge from multiple source models are sufficiently leveraged. (3) A coordinating weight learning method is derived for dynamically revealing the performance differences of different source models over different classes. It is integrated with the semi-supervised bilevel optimization algorithm for consistently updating the teacher and student models. Extensive experiments on five well-known chest radiography datasets show that our approach outperforms state-of-the-art alternatives clearly, along with in-depth ablation studies for verifying the design of our model components.

# 2 Relate Works

**Unsupervised domain adaptation for medical image classification.** There exist shallow UDA and deep UDA approaches in the literature. Shallow UDA approach adapts two routes, *i.e.*, source domain instance weighting [55, 57] and feature transformation [24, 34]. All of these methods need to access source domain data. Similarly, there are also two routes for deep UDA approach. They are domain alignment based [19, 62] and pseudo-labeling based [5]. The first strategy solves the UDA problem by minimizing the domain difference between the source domain and target domain, and is currently the most popular method. Gao *et al*. [19] used the central moment difference matching to perform adaptation of classifying brain MRI data. The second strategy generates dummy data to retrain target model. For multi-label medical image classification, there exists a work based on domain alignment with a multi-label regularization term [46]. Bermúdez Chacón *et al*. [5] used the normalized cross-correlation to generate soft labels for the target domain. The above UDA methods do not update the source domain model, and they are all based on single-source domain. However, the situation of multi-source domains is very common in practical situations.

**Source-free domain adaptation.** Source-free domain adaptation methods can be roughly divided into two routes, *i.e.*, generative approach [27, 28, 33, 63] and pseudo-label approach [4, 26, 35, 56]. The generation approach generates target-style training samples to train the prediction model. Since learning to generate features is difficult, this approach is extremely limited. The pseudo-label approach generates pseudo-labels through the source domain model, which is simple and general and has recently achieved good results in the machine learning community. The research of source-free domain adaptation in the medical image analysis field mainly focuses on image segmentation. Bateson *et al*. [4] maximized the mutual information between the target images and their label predictions to perform spine, prostate and cardiac segmentation. Vibashan *et al*. [56] implemented source-free domain adaptive image segmentation by generating pseudo-labels and applied self-training methods for task-specific representation. These works are all conducted in the single-source domain case. Currently, the research on multi-source-free domain adaptation is extremely limited, and most of the works adapt the method of generating trusted pseudo-labels [1, 14].

**Multi-source domain adaptation for medical image classification.** In machine learning community, MSDA works mainly have two strategies, *i.e.* distribution alignment [43, 74] and adversarial learning [61, 71, 72]. The first strategy computes the statistical discrepancy between multi-source domains and target domain, and then combines all predictions. The second strategy trains a domain discriminator and forces the feature extraction network to learn domain-invariant features to confuse the domain discriminator. For medical image classification, there only exist several shallow DA models. Wang *et al*. [58] proposed to map multiple source and target data to a common latent space for autism spectrum disorder classification. Cheng *et al*. [8] constructed a multi-domain transfer classifier for the early diagnosis of Alzheimer's disease. All of these strategies require to access source domain data and are not suitable for solving the proposed SMDA problem. To the best of our knowledge, current teacher-student domain adaptation methods in the medical and machine learning communities only consider the single-source domain case. When extended to the multi-source domain, it will face a challenging multi-objective optimization problem [10, 41].

**Semi-supervised domain adaptation (SSDA).** Our problem is also related to SSDA which assumes a small number of labeled samples in the target domain. Compared to UDA, using a few labeled samples of the target domain allows to further achieve better domain alignment [31, 44, 66]. Due to the shift of domain distribution, directly applying classical semi-supervised learning methods to the SSDA problem will lead to sub-optimal performance. Representative SSDA works are based on subspace learning [44, 66], entropy minimization [20, 50], label smoothing [13, 48] and active learning [48, 52]. However, all of these methods assume a single source domain with the source domain data accessible. Unlike these works, our method incorporates meta-learning and uses the performance on the labeled target data as a feedback signal.

**Teacher-student domain adaptation models in medical image analysis.** Usually, teacher-student domain adaptation model proposes multiple consistencies to solve UDA problem. To the best of our knowledge, teacher-student based domain adaptation methods have received little attention on medical image analysis. Perone *et al*. [45] proposed a semi-supervised learning based UDA method for medical image segmentation, which minimizes the consistency loss of the predicted results between the student model and the teacher model for unlabeled samples in the target domain during the training process. The network is updated by the exponential moving average of the student

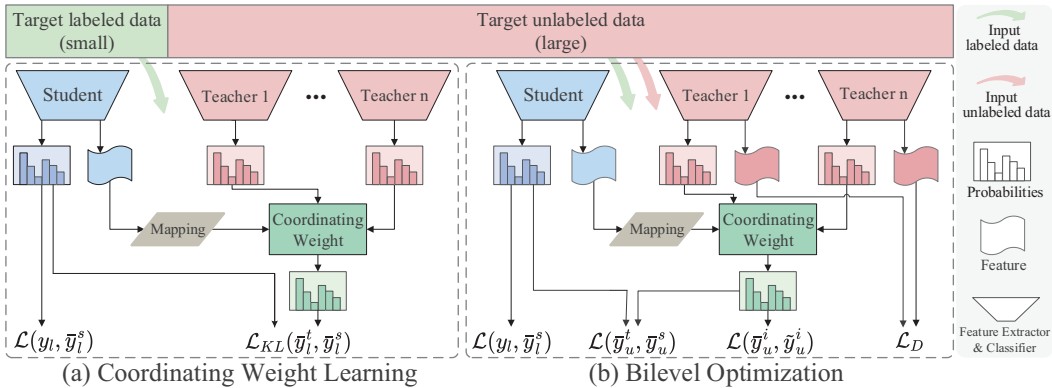

Figure 1: Overview of our proposed MetaTeacher architecture. (a) Learning the coordinating weight mapping which will be used subsequently to provide guidance for updating the teacher models. (b) Alternately updating the teacher and student models. Each teacher is updated with feedback signals from the student and other teachers.

network weights (mean-teacher [54]). The method is experimentally performed on the SCAM (Spinal Cord Anatomy MR Image) dataset to demonstrate its effectiveness. There are several approaches on teacher-student domain adaptation in the field of machine learning. French *et al*. [17] made some modifications to the mean-teacher scheme for the challenging domain adaptation of natural image classification. Cai *et al*. [7] proposed multiple consistency regulations to solve cross-domain detection problem. Deng *et al*. [12] combined the idea of feature alignment and data augmentation based on mean-teacher scheme. These methods all assume single-source domain, and to our knowledge, there is currently no work on multi-source teacher-student domain adaptation. Additionally, the mean-teacher approach does not sufficiently use the feedback signal from the target domain, so the performance improvement is limited.

## 3   Methodology

**Problem statement.** Suppose $D_T = \{(X_L^t, Y_L^t), X_U^t\}$ where $Y_L^t$ denotes label annotations for a small amount of target domain samples $X_L^t$ and $X_U^t$ for target domain samples without any label annotations. The dimension of label vector is $m$. $D_{S_i} = \{(X_L^i, Y_L^i)\}$ where $Y_L^i$ denotes label annotations for $i$-th source domain samples $X_L^i$. For the proposed semi-supervised multi-source-free domain adaptation problem, when the pretrained source classifiers $f_{T_i}$ is applied to the target domain, the source dataset $D_{S_i}$ is not accessible for $i = 1, \cdots, n$. Given the source classifiers $f_{T_i}$ for $i = 1, \cdots, n$ and the target data $D_T$, the *objective* is to find a target-domain mapping $f_S : X_U^t \to Y_U^t$ where $Y_U^t$ denotes the predicted labels for target domain samples $X_U^t$.

**Overview.** As shown in Fig.1, our framework is based on a *multi-teacher and one-student scheme*. First, multiple teacher models are pretrained according to each source domain, and then the student model is initialed using a randomly chosen teacher model. They are all composed of a feature extractor (*e.g.*, Resnet50 [21]) and a multi-label classifier. The classifier consists of a fully connected layer, where the input is an one-dimensional expanded feature, and the output is the probability of each label. The objective function is the error loss between the predicted output and the ground truth.

Compared with traditional teacher-student models, our method is featured with two unique parts: (1) *Coordinating weight learning*; (2) *Bilevel optimization*. For the first part, a mapping is trained based on labeled target domain samples, which fuses the multi-teacher predictions adaptively for each target sample. This mapping will be used in the second part. In the initial iteration, the mapping and student model are trained based on labeled target samples. In the subsequent iterations, this part will only optimize the mapping while the student model will be updated by bilevel optimization. In the bilevel optimization part, the student and teacher models are updated alternately in a meta-learning manner. Specifically, for an unlabeled target sample, a coordinating weight is generated, which provides optimization direction for each teacher model. Finally, these two parts will be iteratively undated until convergence.

## 3.1 Coordinating Weight Learning

As mentioned earlier, the teacher models are trained on different source domain data. Due to different distributions, they often present different characteristics. Therefore, for a target domain sample, the classification probability of each teacher model could be inconsistent. When we want to optimize a teacher model based on the target domain samples, the optimization direction of each teacher model should be different. So it is necessary to obtain the contribution weight of each teacher model to the final classification results. We call this *coordinating weight*. Fortunately, we can obtain the weight mapping with the labeled samples in the target domain.

As shown in Fig.1(a), for obtaining the coordinating weight, we first input the labeled target sample $x_l^t$ into the student network, and get the output $B = g(x_l^t)$ from feature extraction network $g$, where $B \in R^{c \times h \times w}$, with $c$, $h$, and $w$ the number of channels, height, and width respectively. Then, we perform a maximum pool operation on the feature map $B$ to get $\psi \in R^{1 \times c}$ which retains the most important information of each channel. Our mapping consists of two learnable variables $\mu$ and $\nu$, where $\mu \in R^{n \times 1}$, $\nu \in R^{c \times m}$. Then, we define a mapping $\phi = \mu \psi \nu \in R^{n \times m}$ for the target sample $x_l^t$. After normalizing, we get the coordinating weight matrix $W$ where

$$W_{j,k} = \frac{exp(\phi_{j,k})}{\sum_{z=1}^n exp(\phi_{z,k})}. \tag{1}$$

Suppose for the sample $x_l^t$, the predictions of all teachers are formed as a matrix $P \in R^{n \times m}$. By taking the Hadamard product between the teacher predictions and the coordinating weight matrix, we obtain the fused prediction as the following,

$$\bar{y}_l^t = Sum(P \circ W) \tag{2}$$

where $Sum(\cdot)$ means adding by rows. Denoting $\bar{y}_l^s = f_S(x_l^t; \theta_S)$ as the student prediction on the target sample $x_l^t$, we train the weight mapping and initialize student network using the following loss,

$$\mathcal{L}_W = \mathcal{L}\left(\bar{y}_l^s, y_l\right) + \alpha \mathcal{L}_{KL}\left(\bar{y}_l^t, \bar{y}_l^s\right) + \beta\left(\|\mu\| + \|\nu\|\right) \tag{3}$$

where $\mathcal{L}\left(\bar{y}_l^s, y_l\right) = \frac{1}{m}\sum_{i=1}^m [y_{l,i} log(\bar{y}_{l,i}^s) + (1 - y_{l,i}) log(1 - \bar{y}_{l,i}^s)]$ represents the BCE (Binary Cross Entropy) loss, $y_l$ is the ground truth, $\theta_S$ is the parameter of student network. $\mathcal{L}_{KL}\left(\bar{y}_l^t, \bar{y}_l^s\right) = \sum_{i=1}^m \bar{y}_{l,i}^t \log(\bar{y}_{l,i}^t / \bar{y}_{l,i}^s)$ represents the KL (Kullback-Leibler divergence) loss which measures the distribution difference between the fused teacher prediction and student prediction. $\alpha$ and $\beta$ are two balance parameters.

**Remark.** The mapping $\phi$ generates coordinating weight with Eq.(1). It not only reveals the complementarity of different teachers on different instances, but also, more interestingly, participates in the derivation of the update formula of teacher models in the bilevel optimization process (see Appendix), providing a reference for the update direction of different teachers.

## 3.2 Bilevel Optimization

The bilevel optimization problem [6, 9] was first proposed in the field of game theory. It includes an *upper-level optimization task* and a *lower-level optimization task*, where the former contains the latter as a constraint. Here, the upper-level optimization task (student) provides feedback signals to the lower-level optimization tasks (teachers) through the performance on labeled data and the coordinating weight mapping. For an unlabeled target sample $x_u^t$, suppose the pseudo-label based on the learned coordinating weight mapping $\phi$ from multi-teachers Eq.(2) is $\bar{y}_u^t$ and the corresponding coordinating weight matrix is $W_u$, we can define a loss function $\Gamma_u$ as follows,

$$\Gamma_u(\theta_{T_1}, \cdots, \theta_{T_n}, \theta_S) = \mathcal{L}(\bar{y}_u^t, \bar{y}_u^s) \tag{4}$$

where $\bar{y}_u^s = f_S(x_u^t; \theta_S)$, $\theta_{T_i}$ is the parameter of the $i$-th teacher network. Similarly, a loss function $\Gamma_l\left(\theta_{T_1}, \cdots, \theta_{T_n}, \theta_S\right) = \mathcal{L}\left(y_l, \bar{y}_l^s\right)$ is defined for a labeled target samples $x_l^t$. In the bilevel optimization task, updating $\theta_S$ is the upper-level optimization task objective, while updating $\theta_{T_1}, \cdots, \theta_{T_n}$ is the lower-level optimization task objective. The upper-level optimization task and the lower-level optimization task are mutually constrained. To reach the lower-level optimization task objective, the performance of the upper-level optimization task objective on the labeled target data is utilized

as *feedback signal*. We formulate the objective function in lower-level optimization task as the following,

$$\underset{\theta_{T_1}, \cdots, \theta_{T_n}}{\arg\min} \, \Gamma_l\left(\theta_{T_1}, \cdots, \theta_{T_n}, \theta_S^{OP}\right) \quad \text{s.t.} \quad \theta_S^{OP} = \underset{\theta_S}{\arg\min} \, \Gamma_u\left(\theta_{T_1}, \cdots, \theta_{T_n}, \theta_S\right). \tag{5}$$

Eq.(5) cannot be optimized simply by gradient descent algorithm, because the teacher's parameters can not be updated until $\theta_S$ reaches the optimum. To overcome this issue, we resort to the idea of meta-learning [16, 38, 47] by making a one-step approximation of the problem,

$$\theta_S^{OP} \approx \theta_S - \eta_S \cdot \nabla_{\theta_s} \Gamma_u\left(\theta_{T_1}, \theta_{T_2}, \cdots, \theta_{T_n}, \theta_S\right) \tag{6}$$

where $\eta_S$ is the learning rate of the student network. Substituting Eq. (6) into Eq. (5), we obtain a new optimization objective function

$$\Gamma_l\left(\theta_{T_1}, \cdots, \theta_{T_n}, \theta_S - \eta_S \cdot \nabla_{\theta_s} \Gamma_u\left(\theta_{T_1}, \theta_{T_2}, \cdots, \theta_{T_n}, \theta_S\right)\right). \tag{7}$$

By optimizing Eq. (7) (see Appendix), we get the following update rules,

$$\theta_S' = \theta_S - \eta_S \cdot \nabla_{\theta_s} \Gamma_u, \tag{8}$$

$$\theta_{T_i}' = \theta_{T_i} - \eta_{T_i} \cdot \left[(\nabla_{\theta_S'} \Gamma_l)^T \cdot \nabla_{\theta_S} \Gamma_u\right]^T \cdot \nabla_{\theta_{T_i}} \mathcal{L}\left(\bar{y}_u^i, \tilde{y}_u^i\right) \tag{9}$$

for $i = 1, \cdots, n$, where $\theta_S'$ and $\theta_{T_i}'$ are the updated parameters corresponding to the student and teachers respectively. $\bar{y}_u^i = f_{T_i}\left(x_u^t; \theta_{T_i}\right) \cdot W_u^i$ with $W_u^i$ the $i$th-row coordinating weight vector of $W_u$ *w.r.t* the $i$-th teacher. We obtain the pseudo label $\tilde{y}_u^i$ by binarizing $\bar{y}_u^i$ as: $\tilde{y}_{u,j}^i = 0$ when $\bar{y}_{u,j}^i < 0.5$ and $\tilde{y}_{u,j}^i = 1$ for the other cases.

Additionally, in order to prevent optimizing teachers in the same direction, the predictions of the updated multiple teachers should be as far away from each other as possible. To that end, we define a divergence loss as follows,

$$\mathcal{L}_D = -ln \sum_{j=1, j \neq i}^{n} \mathcal{L}_2\left(B_{T_i}\left(x_u^t; \theta_{T_i}\right), B_{T_j}\left(x_u^t; \theta_{T_j}\right)\right) \tag{10}$$

where $B_{T_i}\left(x_u; \theta_{T_i}\right)$ represents the max-pooled results of the output feature map of the $i$-th teacher network. Here, we apply a max-pooling operation to the output features of multiple teachers and calculate the distance with $L_2$ norm. By requiring these feature maps to be far away each other, the optimization direction of teachers will be effectively adjusted. Finally, we update the $i$-th teacher network by the following rule,

$$\theta_{T_i}' = \theta_{T_i} - \eta_{T_i} \cdot \left(\left[(\nabla_{\theta_{S'}} \Gamma_l)^T \cdot \nabla_{\theta_S} \Gamma_u\right]^T \cdot \nabla_{\theta_{T_i}} \mathcal{L}\left(\bar{y}_u^i, \tilde{y}_u^i\right) + \gamma \nabla_{\theta_{T_i}} \mathcal{L}_D\right) \tag{11}$$

where $\gamma$ is a hyperparameter.

**Remark.** Eq.(11) reveals that the update direction of $\theta_{T_i}$ is determined by three factors: (1) Coordinating weight confuses the feedback signals from different teachers; (2) Student network parameters provide the feedback signals and generate coordinating weight; (3) Diversity constraint emphasizes the characteristic of different teacher networks. Interestingly, these three factors change over time during the meta-learning process. In addition to alternating updates of the student and teacher models, we also update the mapping periodically.

### 3.3 Optimization Process

The optimization process alternates between upper-level optimization and lower-level optimization. Since the coordinating weight mapping is fixed in the process of bilevel optimization, with the progress of meta-learning, the coordinating weight $W_u$ can not reflect the internal relationship between the source domains and the target domain gradually. Therefore, we use the learned teachers to update the mapping at regular intervals. The training process is summarized in Algorithm 1.

## 4 Experiments

**Datasets.** Five publicly available chest x-ray datasets are used to construct our multi-domain adaptation scenarios. *NIH-CXR14* [59] is a large public dataset of chest x-ray, which contains 108,948

---
**Algorithm 1** Our proposed MetaTeacher method.
---
**Require:** Student network parameters $S^{(0)}$, teacher network parameters $T_1^{(0)} \sim T_n^{(0)}$, labeled data $(x_l^t, y_l)$, unlabeled data $(x_u^t)$, hyperparameters $\alpha, \beta, \gamma$, mapping updating interval $\mathcal{T}$.
**Ensure:** Optimized student model $S^{(N)}$.
  1: **function** METATEACHER($S^{(0)}, T_1^{(0)} \sim T_n^{(0)}, \alpha, \beta, \gamma, \mathcal{T}$)
  2:   $S^{(0)}$, mapping $\leftarrow$ *Coordinating Weight Learning*
  3:   **for** $t = 0 \rightarrow N - 1$ **do**
       **Upper-level optimization:**
  4:     Compute gradient $\nabla_{\theta_{S^{(t)}}} \mathcal{L}_u$
  5:     Update the student: $\theta_{S^{(t+1)}} \leftarrow \theta_{S^{(t)}} - \eta_S \nabla_{\theta_{S^t}} \mathcal{L}_u$                          $\triangleright$ Eq.(8)
       **Lower-level optimization:**
  6:     Compute gradient $\nabla_{\theta_{S^{(t+1)}}} \mathcal{L}_l$
  7:     **for all** $T_1^{(t)} \sim T_n^{(t)}$ **do**
  8:       Compute gradient $\nabla_{\theta_{T_i^{(t)}}} \mathcal{L}\left(\bar{y}_u^i, \tilde{y}_u^i\right)$
  9:       Compute gradient $\nabla_{\theta_{T_i^{(t)}}} \mathcal{L}_D$                          $\triangleright$ Eq.(10)
 10:       Update the $i$-th teacher: $\theta_{T_i^{(t+1)}} \leftarrow \theta_{T_i^{(t)}} - \eta_{T_i} \cdot$                          $\triangleright$ Eq.(11)
             $([(\nabla_{\theta_{S^{(t+1)}}} \Gamma_l)^T \cdot \nabla_{\theta_{S^{(t)}}} \Gamma_u]^T \cdot \nabla_{\theta_{T_i^{(t)}}} \mathcal{L}\left(\bar{y}_u^i, \tilde{y}_u^i\right) + \gamma \nabla_{\theta_{T_i^{(t)}}} \mathcal{L}_D)$
 11:     **end for**
 12:     **if** $t \% \mathcal{T} = 0$ **then**
 13:       mapping $\leftarrow$ *Coordinating Weight Learning*
 14:     **end if**
 15:   **end for**
 16:   **return** $S^{(N)}$
 17: **end function**
---

front view x-ray images of 32,717 patients collected from NIH Clinical Center, with a total of 14 disease labels. *MIMIC-CXR* [23] contains 377,110 images and text reports, corresponding to 227,835 radiological studies conducted by Beth Israel Deaconess Medical Center in Boston, Massachusetts. *CheXpert* [22] consists of 224,316 chest x-ray of 65,240 patients. The dataset collected chest x-ray examinations and related radiology reports performed at inpatient and outpatient centers at Stanford Hospital from October 2002 to July 2017. *Open-i* [11] is collected by Indiana University Hospital through the network from open source literature and biomedical image collection. It contains 3955 radiology reports, corresponding to 7470 frontal and lateral chest films. To be consistent with other datasets, we filter out the side chest x-ray in Open-I, leaving only 3955 frontal images. *Google-Health-CXR* [3] is manually labeled by medical experts for CXR images with high accuracy and contains about 4000 images. We follow the traditional UDA setting, and choose the disease closed set in these five datasets as multi classification labels, *i.e.*, Atelectasis, Cardiomegaly, Effusion, Consolidation, Edema and Pneumonia. Four transfer scenarios are constructed, which are *NIH-CXR14, CheXpert, MIMIC-CXR to Open-i*; *NIH-CXR14, CheXpert, MIMIC-CXR to Google-Health-CXR*; *CheXpert, MIMIC-CXR to NIH-CXR14* and *NIH-CXR14, CheXpert to Open-i*.

**Implementation details.** In order to make a compromise between images in different datasets, we scale the images to 128*128 before feeding them into the network. To expand the training set, several data augmentation techniques are used, including random cropping and horizontal flipping. SGD with momentum of 0.9 is used as the optimizer. For the student model, the initial learning rate is 0.01 and the weight decay is 5e-4. The learning rate for coordinating weight mapping is 0.001; For the teacher models, the initial learning rate is 0.001 and the weight decay is 5e-6. The values of $\alpha$, $\beta$ and $\gamma$ are set as 0.5, 0.01 and 0.01 respectively. For the case when the target domains datasets are small-scale, such as *Open-i* and *Google-Health-CXR*, we assume that there are 200 labeled data in the target domains, and in order to give a good initial condition for training, we randomly select a source model to initialize the target model. For the case when the target domains datasets are large-scale, such as *NIH-CXR14*, we assume that there are 500 labeled data in the target domains. Unless otherwise specified, the interval for updating coordinating weight mapping is set as 100 iterations. Following the setting of multi-label medical image classification problems, the evaluation criterion is Area Under the Receiver Operating Characteristic (AUROC) [15] curve score.

Table 1: Comparing the state-of-the-art methods on the transfer from *NIH-CXR14, CheXpert, MIMIC-CXR* to *Open-i*. Metric: AUROC.

| Method | Atelectasis | Cardiomegaly | Effusion | Consolidation | Edema | Pneumonia | *Average* |
|---|---|---|---|---|---|---|---|
| DECISION [1] | **83.27** | 91.55 | 96.18 | 97.02 | 92.74 | 89.24 | 91.67 |
| CAiDA [14] | 82.45 | 92.16 | 95.12 | 95.92 | 89.89 | 90.37 | 90.99 |
| SHOT-best [35] | 81.48 | 91.22 | 94.19 | 95.10 | 88.96 | 89.58 | 90.09 |
| MME [50] | 82.44 | 90.82 | 95.46 | 96.07 | 90.26 | 87.20 | 90.38 |
| ECACL [30] | 82.60 | 92.18 | 96.32 | 95.97 | 90.70 | 89.61 | 91.23 |
| Source Only(N) | 83.09 | 87.20 | 96.11 | 95.10 | 86.87 | 77.40 | 87.63 |
| Source Only(C) | 82.26 | 87.64 | 94.71 | 96.61 | 90.22 | 75.12 | 87.76 |
| Source Only(M) | 80.63 | 91.31 | 94.87 | 94.53 | 84.91 | 82.78 | 88.05 |
| Fine-tune(*average*) | 82.14 | 88.71 | 95.32 | 95.52 | 88.77 | 78.48 | 88.16 |
| MetaTeacher(w/o *mapping*) | 79.99 | **92.64** | **98.22** | 93.64 | **95.50** | 84.54 | 90.76 |
| MetaTeacher(w/o *update*) | 81.98 | 90.72 | 95.76 | 95.51 | 89.40 | 82.53 | 89.32 |
| MetaTeacher(*all*) | 81.72 | 92.59 | 96.25 | **97.64** | 94.52 | **94.33** | **92.84** |

Table 2: Comparing the state-of-the-art methods on the transfer from *NIH-CXR14, CheXpert, MIMIC-CXR* to *Google-Health-CXR*. Metric: AUROC.

| Method | Atelectasis | Cardiomegaly | Effusion | Consolidation | Edema | Pneumonia | *Average* |
|---|---|---|---|---|---|---|---|
| DECISION [1] | 77.24 | 81.71 | 85.94 | 79.03 | 83.48 | 83.68 | 81.85 |
| CAiDA [14] | 76.90 | 81.82 | 87.55 | 79.62 | 85.10 | 82.72 | 82.29 |
| SHOT-best [35] | 75.43 | 80.28 | 86.63 | 77.88 | 82.37 | 81.22 | 80.64 |
| MME [50] | 77.34 | **84.93** | 86.17 | 78.65 | 85.33 | 71.28 | 80.62 |
| ECACL [30] | 76.27 | 84.54 | 87.06 | 79.95 | 85.82 | 72.66 | 81.05 |
| Source Only(N) | 76.54 | 84.48 | 86.36 | 75.66 | 83.94 | 62.59 | 78.26 |
| Source Only(C) | 72.09 | 76.45 | 84.55 | 79.07 | 68.25 | 58.39 | 73.13 |
| Source Only(M) | 68.04 | 79.38 | 84.17 | 72.41 | 68.71 | 52.60 | 70.88 |
| Fine-tune(*average*) | 73.48 | 80.14 | 85.96 | 74.17 | 74.74 | 60.20 | 74.78 |
| MetaTeacher(w/o *mapping*) | 75.62 | 83.91 | 85.40 | **80.27** | 75.13 | 81.77 | 80.35 |
| MetaTeacher(w/o *update*) | 76.75 | 84.30 | 86.67 | 78.59 | 82.31 | 65.84 | 79.08 |
| MetaTeacher(all) | **77.65** | 79.52 | **88.73** | 78.74 | **86.73** | **84.78** | **82.69** |

## 4.1 Comparisons to State-of-the-Art

At present, there does not exist any experimental report on our problem setting. So we choose four category of methods for compare. The first category is Source only which means directly applying a teacher model to the target domain. The second category is Fine-tune(*average*) which fine-tune each teacher network using labeled target domain data, then average their predicted values. The third category is the state-of-the-art multi-source-free domain adaptation methods, which are DECISION [1], CAiDA [14], and SHOT-best. The SHOT-best refers to adapting each source domain separately through the SHOT [35] method. The model with the best performance on the validation set is selected. The final category is semi-supervised domain adaptation methods, including MME [50] and ECACL [30]. For the semi-supervised domain adaptation methods, we assume that the labeled target data are the same as our method. Since they are single-source based methods, we perform domain adaptation for each source model and take the best result.

Tables 1-4 show the comparison results on four transfer scenarios. MetaTeacher(all) is our proposed method. Source Only(N), Source Only(C) and Source Only(M) are the teacher models respect to the *NIH-CXR14*, *CheXpert* and *MIMIC-CXR* datasets respectively. For the scenario from *CheXpert*, *MIMIC-CXR* to *NIH-CXR14*, since the dataset *NIH-CXR14* contains 108,948 x-ray images, different from other scenarios, this time we do not need to initialize the target model with the source models. It can be observed that our method achieves the best performance. The extensive experiments on four different transfer scenarios verify the adaptability of our method under multi-label chest x-ray dataset transfer cases. For the scenario from *NIH-CXR14*, *CheXpert* to *Open-i*, as show in Table 4, the

Table 3: Comparing the state-of-the-art methods on the transfer from *CheXpert, MIMIC-CXR* to *NIH-CXR14*. Metric: AUROC .

| Method | Atelectasis | Cardiomegaly | Effusion | Consolidation | Edema | Pneumonia | Average |
|---|---|---|---|---|---|---|---|
| DECISION [1] | 72.99 | 80.73 | 79.37 | **75.52** | 82.30 | 71.38 | 77.05 |
| CAiDA [14] | 72.64 | 81.12 | 80.25 | 74.73 | 81.02 | 70.44 | 76.70 |
| SHOT-best [35] | 70.79 | 79.62 | 79.24 | 72.25 | 80.79 | 69.65 | 75.39 |
| MME [50] | 72.90 | 81.73 | 81.01 | 73.11 | 81.03 | 71.52 | 76.88 |
| ECACL [30] | 72.41 | 81.98 | **82.07** | 72.92 | 80.82 | **71.65** | 76.98 |
| Source Only(N) | 72.31 | 80.52 | 79.42 | 69.66 | 77.95 | 67.37 | 74.54 |
| Source Only(C) | 70.45 | 79.66 | 79.98 | 68.26 | 78.01 | 70.82 | 73.86 |
| Fine-tune(*average*) | 71.52 | 80.29 | 80.08 | 68.97 | 78.02 | 69.05 | 74.66 |
| MetaTeacher(w/o *mapping*) | 72.05 | 81.58 | 78.36 | 72.94 | 82.19 | 69.82 | 76.16 |
| MetaTeacher(w/o *update*) | 72.24 | 80.69 | 79.56 | 69.80 | 78.13 | 70.55 | 75.16 |
| MetaTeacher(all) | **73.63** | **86.64** | 80.86 | 72.24 | **86.68** | 66.37 | **77.74** |

Table 4: Comparing the state-of-the-art methods on the transfer from *NIH-CXR14, CheXpert* to *Open-i*. Metric: AUROC.

| Method | Atelectasis | Cardiomegaly | Effusion | Consolidation | Edema | Pneumonia | *Average* |
|---|---|---|---|---|---|---|---|
| DECISION [1] | 83.15 | 90.86 | 96.12 | 96.32 | 92.33 | 88.79 | 91.26 |
| CAiDA [14] | 82.38 | 91.97 | 94.89 | 95.30 | 89.81 | 90.44 | 90.80 |
| SHOT-best [35] | 81.48 | 91.22 | 94.19 | 95.10 | 88.96 | 89.58 | 90.09 |
| MME [50] | 81.46 | 90.40 | 94.86 | **97.73** | 89.79 | 87.31 | 90.26 |
| ECACL [30] | 82.22 | 88.76 | 96.04 | 96.85 | **92.43** | 87.90 | 90.70 |
| Source Only(N) | 83.09 | 87.20 | 96.11 | 95.10 | 86.87 | 77.40 | 87.63 |
| Source Only(C) | 82.26 | 87.64 | 94.71 | 96.61 | 90.22 | 75.12 | 87.76 |
| Fine-tune(*average*) | 82.66 | 87.98 | 95.85 | 95.67 | 88.58 | 77.02 | 87.96 |
| MetaTeacher(w/o *mapping*) | **83.73** | **93.37** | 96.04 | 97.30 | 91.51 | 82.34 | 90.72 |
| MetaTeacher(w/o *update*) | 82.70 | 88.91 | 95.47 | 95.48 | 88.96 | 78.85 | 88.40 |
| MetaTeacher(all) | 82.11 | 92.42 | **96.80** | 97.07 | **92.20** | **91.27** | **91.98** |

performance of two source domains is 0.86% lower than that of three source domains. Furthermore, MetaTeacher also has moderate training time and more clearer background (see Appendix).

## 4.2 Ablation Analysis and Discussion

**Component analysis.** In Tables 1-4, MetaTeacher(w/o *mapping*) represents that our proposed method removes the part of coordinating weight learning and optimization substituted by average. MetaTeacher(w/o *update*) means to remove the bilevel optimization process. In this situation, the weighted output of teachers is used to supervise the learning of student network. The results in the last three rows of Tables 1-4 show that these two parts are indispensable. It is worth mentioning that MetaTeacher(w/o *mapping*) still obtains promising performance due to the following reasons. First, for student updating, averaging predictions from multiple teachers is beneficial for student performance, consistent with the finding by [68]. Second, the fixed $W$ is also involved in the teacher optimization. It means bilevel optimization contributes more gain to the overall performance than the coordinating weight learning. However, the coordinating weight learning can judge which disease category the teacher is good at by weight, knowledge with different weights can be learned from different teachers. Therefore, the results in each disease category are close to the predictions of the best teachers, such as Pneumonia in Table 1 and Atelectasis in Table 2 (also see Appendix).

**Effects of proportion of labeled target data.** Table 5 shows the influence of the amount of labeled data in the target domain on the transfer scenario of *NIH-CXR14, CheXpert, MIMIC-CXR* to *Open-i*. The experimental results show that the performance slowly improves as the amount of labeled data increases; a small number of labeled target domain samples can achieve good results.

Table 5: Effect of the size of labeled target data on the transfer from *NIH-CXR14, CheXpert, MIMIC-CXR* to *Open-i*. Metric: AUROC.

| Number(propotion) | Atelectasis | Cardiomegaly | Effusion | Consolidation | Edema | Pneumonia | *Average* |
|---|---|---|---|---|---|---|---|
| 50(1.4%) | 82.48 | 92.22 | 95.19 | 96.10 | 89.96 | 90.58 | 91.09 |
| 100(2.8%) | 82.19 | 92.50 | 96.83 | 97.02 | 92.43 | 91.20 | 92.03 |
| 200(5.6%) | 81.72 | 92.59 | 96.25 | 97.64 | 94.52 | 94.33 | 92.84 |
| 300(8.4%) | 82.21 | 92.97 | 96.83 | 97.42 | 94.07 | 94.33 | 92.97 |

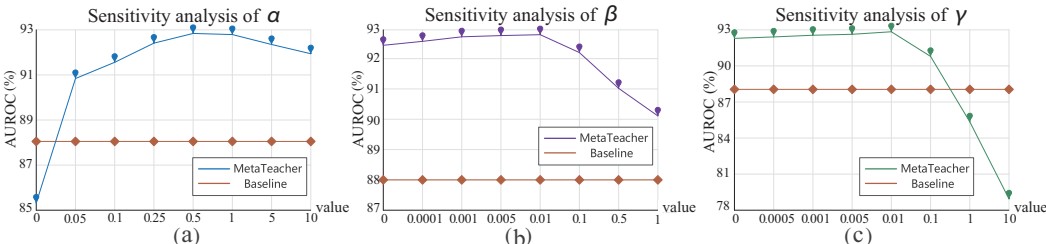

Figure 2: Effect of different hyperparameters on the transfer from *NIH-CXR14, CheXpert, MIMIC-CXR* to *Open-i*. Baseline: source only(M).

**Parameter analysis.** We conduct parameter analysis experiment on the transfer scenario of *NIH-CXR14, CheXpert, MIMIC-CXR* to *Open-i*. The basic strategy is to change a parameter while other parameters are fixed. Our method MetaTeacher has three hyperparameters, *i.e.*, $\alpha$ and $\beta$ in Eq. (3), and $\gamma$ in Eq.(11). Fig.(2)(a) shows performance changing with the parameter $\alpha$. When $\alpha = 0$, the coordinating weight mapping is not trained effectively resulting in the inability to determine the optimization direction of each teacher. When $\alpha$ gradually increases to around 0.5, the result achieve optimal performance. Fig.(2)(b) shows the influences of the parameters $\beta$. When the $\beta$ is too large, it means that the coordinating weight learning part is ineffective and cannot express the relationship between the source domains. When $\beta$ is set to 0, coordinating weight learning may overfit, which may cause coordinating weights to work well on some instances but poorly on other instances; for this case, the performance is 92.49% about 0.35% lower than the result 92.84% in Table 1. Fig.(2)(c) shows the influences of the parameter $\gamma$ on divergence loss. When $\gamma$ is set to 0.01, the performance reaches the best, but with the continuous increase of $\gamma$, the performance decreases obviously. When $\gamma = 0$, the result is 92.29%, which is 0.55% lower. We can also see that our method is also quite stable for the parameters $\alpha, \beta$ and $\gamma$ in a large interval.

## 5 Conclusion

In this paper, we have proposed a novel framework, termed as MetaTeacher, for semi-supervised multi-source-free domain adaptation for medical image classification. The transfer learning process is modeled as a multi-teacher and one-student scheme. We not only optimize the student, but also optimize the teachers through the student's feedback in the target domain. Our optimization is based on meta-learning with two main parts: coordinating weight learning, and bilevel optimization. The first part obtains the coordinating weight mapping which is then used to coordinate the teacher outputs and updates. Bilevel optimization updates the student based on the pseudo-labeled data produced by the teachers and updates each teacher based on the feedback signal generated by the student and other teachers. Extensive experiments on multi-label chest x-ray datasets empirically demonstrated the superiority of our method over many state-of-the-art approaches.

## Acknowledgments and Disclosure of Funding

This work was supported in part by the National Key R&D Program of China (2018YFE0203900), National Natural Science Foundation of China (62276048), Sichuan Science and Technology Program (2020YFG0476).

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
