# MetaTeacher: Coordinating Multi-Model Domain Adaptation for Medical Image Classification (Appendix)

**Zhenbin Wang**[1], **Mao Ye**[1]*, **Xiatian Zhu**[2], **Liuhan Peng**[3], **Liang Tian**[1], **Yingying Zhu**[4]

[1]University of Electronic Science and Technology of China, Chengdu, China
[2]University of Surrey, Guildford, UK
[3]Xinjiang University, Ürümqi, China
[4]University of Texas, Arlington, US
*zhenbinwang@foxmail.com, cvlab.uestc@gmail.com, xiatian.zhu@surrey.ac.uk*

## A    Updating Rules of Bilevel Optimization

We follow the derivation route in  [7] except the coordinating weight part. For expression clarity, let $|T_i|$ and $|S|$ denote the dimensions of $\theta_{T_i}$ and $\theta_S$ respectively, where $\theta_S \in R^{|S| \times 1}$ and $\theta_{T_i} \in R^{|T_i| \times 1}$. Suppose there is a batch of unlabeled target samples $x_u^t$, the $j$-th teacher $T_j$ samples the pseudo labels $\hat{y}_u^j \sim f_{T_i}(x_u^t; \theta_{T_j})$ for $j = 1, \cdots, n$ and

$$\bar{y}_u^t = \sum_{j=1}^n \bar{y}_u^j = \sum_{j=1}^n \hat{y}_u^j \circ W_u^j \tag{12}$$

where $W_u^j$ is the $j$-th row of coordinating weight matrix $W_u$ *w.r.t* $x_u^t$ and $\circ$ is the Hadamard product. So we can use $(x_u^t, \bar{y}_u^t)$ to update the parameter $\theta_S$ according to Eq.(6) in expectation as follows,

$$\theta_S' = E_{\bar{y}_u^t} \left[ \theta_S - \eta_S \cdot \nabla_{\theta_s} \mathcal{L} \left( \bar{y}_u^t, f_S \left( x_u^t; \theta_S \right) \right) \right]. \tag{13}$$

According to Eq.(7), we update $\theta_{T_i}$ based on a batch of labeled target data $(x_l^t, y_l)$ by optimization the following objective function

$$\underset{\theta_{T_1}, \cdots, \theta_{T_n}}{\arg\min} \mathcal{L} \left( y_l, f_S \left( x_l^t, \theta_S' \right) \right). \tag{14}$$

For end-to-end optimization with gradient descent, the partial derivative respect to the above objective function $R$ is

$$\underbrace{\frac{\partial R}{\theta_{T_i}}}_{1 \times |T_i|} = \frac{\partial \mathcal{L} \left( y_l, f_S \left( x_l^t; \theta_S' \right) \right)}{\partial \theta_{T_i}}, \tag{15}$$

for $i = 1, \cdots n$. According to the chain rule, Eq.(15) can be written as:

$$\underbrace{\frac{\partial R}{\theta_{T_i}}}_{1 \times |T_i|} = \underbrace{\frac{\partial \mathcal{L} \left( y_l, f_S \left( x_l^t; \theta_S' \right) \right)}{\partial \theta_S'}}_{1 \times |S|} \cdot \underbrace{\frac{\partial \theta_S'}{\partial \theta_{T_i}}}_{|S| \times |T_i|}. \tag{16}$$

For the right part of Eq.(16), it follows that

$$\underbrace{\frac{\partial \theta_S'}{\partial \theta_{T_i}}}_{|S| \times |T_i|} = \underbrace{\frac{\partial E \left[ \theta_S - \eta_S \cdot \left( \frac{\partial \mathcal{L} \left( \bar{y}_u^t, f_S(x_u^t; \theta_S) \right)}{\partial \theta_S} \right)^\mathsf{T} \right]}{\partial \theta_{T_i}}}_{|S| \times |T_i|} = \underbrace{\frac{\partial E \left[ -\eta_S \cdot \left( \frac{\partial \mathcal{L} \left( \bar{y}_u^t, f_S(x_u^t; \theta_S) \right)}{\partial \theta_S} \right)^\mathsf{T} \right]}{\partial \theta_{T_i}}}_{|S| \times |T_i|} \tag{17}$$

---

*The corresponding author.

36th Conference on Neural Information Processing Systems (NeurIPS 2022).

where $(\cdot)^T$ is the transpose notation. Suppose that

$$\underbrace{G(\theta_S, \bar{y}_u^t)}_{|S| \times 1} = \underbrace{\left( \frac{\partial \mathcal{L}\left( \bar{y}_u^t, f_S\left(x_u^t; \theta_S\right) \right)}{\partial \theta_S} \right)^{\mathsf{T}}}_{|S| \times 1}, \tag{18}$$

it follows that

$$
\begin{aligned}
\underbrace{\frac{\partial \theta_S'}{\partial \theta_{T_i}}}_{|S| \times |T_i|} &= -\eta_S \cdot \underbrace{\frac{\partial E\left[ G\left(\theta_S, \bar{y}_u^t\right)\right]}{\partial \theta_{T_i}}}_{|S| \times |T_i|} \\
&= -\eta_S \cdot \sum \underbrace{\frac{\partial \left[ G\left(\theta_S, \bar{y}_u^t\right) \cdot P\left( \bar{y}_u^t \mid x_u^t : \theta_{T_1}, \theta_{T_2}, \cdots, \theta_{T_n}, W_u\right)\right]}{\partial \theta_{T_i}}}_{|S| \times |T_i|} \\
&= -\eta_S \cdot \sum \underbrace{G\left(\theta_S, \bar{y}_u^t\right)}_{|S| \times 1} \cdot \underbrace{\frac{\partial P\left( \bar{y}_u^t \mid x_u^t : \theta_{T_1}, \theta_{T_2}, \cdots, \theta_{T_n}, W_u\right)}{\partial \theta_{T_i}}}_{1 \times |T_i|} \, .
\end{aligned}
\tag{19}
$$

Since $\bar{y}_u^t = \sum_{j=1}^{n} \bar{y}_u^j = \sum_{j=1}^{n} \hat{y}_u^j \circ W_u^j$, Eq.(19) can be further resolved

$$
\begin{aligned}
\underbrace{\frac{\partial \theta_S'}{\partial \theta_{T_i}}}_{|S| \times |T_i|} &= -\eta_S \cdot \sum \underbrace{G\left(\theta_S, \bar{y}_u^t\right)}_{|S| \times 1} \cdot \underbrace{\frac{\partial \left( \sum_{j=1}^{n} P\left( \bar{y}_u^j \mid x_u^t; \theta_{T_j}, W_u^j\right) \right)}{\partial \theta_{T_i}}}_{1 \times |T_i|} \\
&= -\eta_S \cdot \sum \underbrace{G\left(\theta_S, \bar{y}_u^t\right)}_{|S| \times 1} \cdot \underbrace{\frac{\partial \left( P\left( \bar{y}_u^i \mid x_u^t; \theta_{T_i}, W_u^i\right) \right)}{\partial \theta_{T_i}}}_{1 \times |T_i|} \, .
\end{aligned}
\tag{20}
$$

From the REINFORCE equation [9], we can get

$$
\begin{aligned}
\sum & \underbrace{\frac{\partial \left( P\left( \bar{y}_u^i \mid x_u^t; \theta_{T_i}, W_u^i\right) \right)}{\partial \theta_{T_i}}}_{1 \times |T_i|} \\
&= \sum \left( P\left( \bar{y}_u^i \mid x_u^t; \theta_{T_i}, W_u^i\right) \right) \cdot \underbrace{\frac{\partial \log \left( P\left( \bar{y}_u^i \mid x_u^t; \theta_{T_i}, W_u^i\right) \right)}{\partial \theta_{T_i}}}_{1 \times |T_i|} \\
&= -E \underbrace{\left[ \frac{\partial \mathcal{L}\left( \left( f_{T_i}\left(x_u^t; \theta_{T_i}\right) \cdot W_u^i\right), \tilde{y}_u^i\right)}{\partial \theta_{T_i}} \right]}_{1 \times |T_i|}
\end{aligned}
\tag{21}
$$

where $\tilde{y}_u^i$ is the pseudo labels after normalizing the values of $f_{T_i}\left(x_u^t; \theta_{T_i}\right) \cdot W_u^i$ to 0 or 1, *i.e.*, $\tilde{y}_{u,j}^i = 0$ when $\hat{y}_{u,j}^i < 0.5$ and $\tilde{y}_{u,j}^i = 1$ for other cases. After substituting Eq.(21) into Eq.(19), Eq. (18) into Eq.(19), and Eq.(19) into Eq.(16), we obtain the following gradient,

$$\underbrace{\frac{\partial R}{\theta_{T_i}}}_{1 \times |T_i|} = \eta_S \cdot \underbrace{\frac{\partial \mathcal{L}\left( y_l, f_S\left(x_l^t; \theta_S'\right)\right)}{\partial \theta_S'}}_{1 \times |S|} \cdot E \left[ \underbrace{G\left(\theta_S, \bar{y}_u^t\right)}_{|S| \times 1} \cdot \underbrace{\frac{\partial \mathcal{L}\left( f_{T_i}\left(x_u^t; \theta_{T_i}\right) \cdot W_u^i, \tilde{y}_u^i\right)}{\partial \theta_{T_i}}}_{1 \times |T_i|} \right]. \tag{22}$$

By Monte Carlo approximation, we use the sampled $\hat{y}_u^j$ for $j = 1, \cdots, n$ to obtain the update rules,

$$\theta_S' = \theta_S - \eta_S \cdot \nabla_{\theta_s} \Gamma_u \tag{23}$$

$$
\begin{aligned}
\theta'_{T_i} &= \theta_{T_i} - \eta_{T_i} \cdot \underbrace{\frac{\partial \mathcal{L}\left(y_l, f_S\left(x_l^t; \theta'_S\right)\right)}{\partial \theta_{S'}}}_{1 \times |S|} \cdot \underbrace{\left(\frac{\partial \mathcal{L}\left(\bar{y}_u^t, f_S\left(x_u^t; \theta_S\right)\right)}{\partial \theta_S}\right)^{\mathsf{T}}}_{|S| \times 1} \cdot \underbrace{\frac{\partial \mathcal{L}\left(f_{T_i}\left(x_u^t; \theta_{T_i}\right) \cdot W_u^i, \tilde{y}_u^i\right)}{\partial \theta_{T_i}}}_{1 \times |T_i|} \\
&= \theta_{T_i} - \eta_{T_i} \cdot \left[\left(\nabla_{\theta_{S'}} \Gamma_l\right)^{\mathsf{T}} \cdot \nabla_{\theta_S} \Gamma_u\right]^{\mathsf{T}} \cdot \nabla_{\theta_{T_i}} \mathcal{L}\left(f_{T_i}\left(x_u^t; \theta_{T_i}\right) \cdot W_u^i, \tilde{y}_u^i\right).
\end{aligned}
\tag{24}
$$

# B  Visualization

## B.1  Visualization of ablation analysis

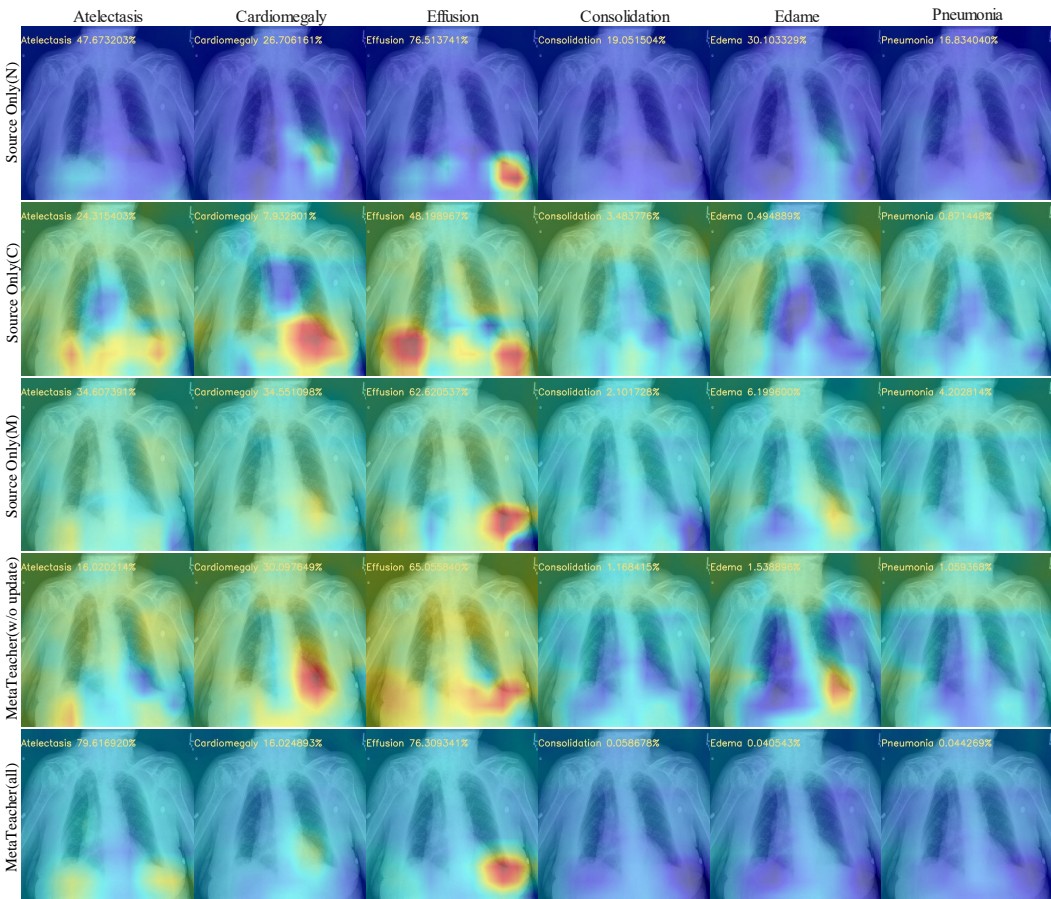

Figure 3: The Class Activation Map (CAM) [10] is used to perform visual ablation analysis on a chest x-ray image in *Open-i* dataset. The background color is blue, with red or yellow representing the disease location. The number on the top left corner of each image is the predicted probability for the corresponding disease. *Zoom in for best view.*

We visualize the domain adaptation performance on the transfer scenario *NIH-CXR14, CheXpert, MIMIC-CXR* to *Open-i*. The visualization sample in the *Open-i* is suffering from Atelecsis and Effusion disease. The comparison models are 1) Source only(N): the model trained on the *NIH-CXR14* dataset; 2) Source only(C): the model trained on the *Chexpert* dataset; 3) Source only(M): the model trained on *MIMIC-CXR* dataset; 4) MetaTeacher(*w/o update*): our approach only containing coordinating weight learning part; 5) MetaTeacher(*all*): our full approach MetaTeacher containing both coordinating weight learning and bilevel optimization.

From the visualization results, we have the following observations. 1) The source models trained on different datasets have different concerns about different diseases. It can be seen that Source only(N) and Source only(M) can identify patients with Effusion disease, with probabilities of 76.513741% and 62.620537%, respectively. However, Source only(C) shows that the patient has only a 48.198967% probability of Effusion disease. 2) Simply fusing multiple teacher predictions does not work in the target domain. MetaTeacher(*w/o update*) is a distillation learning with coordinating weights, which can coordinate the knowledge of each teacher about each disease category. As shown in Fig.3, if most of the source models can detect some disease, the fused model can also detect this disease, and its probability is slightly lower than the maximum value such as Effusion disease. Conversely, if the disease cannot be detected by most of the source models, the fused model can not detected it too such as Atelectasis, Consolidation and Edame. 3) Collaboratively updating teacher and student models works in the target domain. MetaTeacher(*all*) can learn knowledge that the source model does not have. None of these three source domain models can accurately detect the Atelectasis disease, but MetaTeacher(*all*) can identify it, and the output probability is as high as 79.616920%. In addition, for Consolidation, Edame and Pneumonia diseases, MetaTeacher(*all*) predictions for them are close to 0, which shows that our method has a more clear judgment ability for non-existing diseases.

### B.2   Visualization of different methods

The comparison models are DECISION [1] and MME [8]. The first method is a multi-source-free domain adaptation approach, which works by learning a set of weight values corresponding to each source domain model, while learning these weights by using unlabeled target data, then combining the predictions from each source domain as the final prediction. To fit the problem setting, the performance of DECISION is visualized on the transfer scenario *NIH-CXR14, CheXpert,MIMIC-CXR* to *Open-i*. The second method is a single-source semi-supervised domain adaptation approach, which alternately maximizing the conditional entropy of unlabeled target data with respect to the classifier and minimizing it with respect to the feature encoder. Similarly to fit the problem setting, we visualize the MME performance on the transfer scenario *MIMIC-CXR* to *Open-i*. The visualization sample in Fig.4(a) is suffering from Atelecsis and Effusion disease while the sample in Fig.4(b) is suffering from Cardiomegaly disease.

As shown in Fig.4, both of MME and DECISION cannot detect the corresponding diseases. From the visualization results, it can be seen that MME and DECISION contain a widely distributed yellow color, mixed with the red part, which affects their judgments of the disease. For example, Atelectasis and Effusion diseases in Fig.4(a), or Cardiomegaly disease in Fig.4(b), although MME and DECISION can mark the disease location in red, they also contain a lot of yellow color in other places, which confuse their attentions to the right diseases. Unlike them, MetaTeacher contains more blue background color, which can more clearly distinguish the background color from the disease location. Therefore, the disease can be judged more accurately. Additionally, for diseases that are clearly not present in the figure, such as Consolidation, Edema and Pneumonia diseases in Fig.4(a), or Atelectasis and Pneumonia in Fig.4(b). The widespread yellow color makes MME and DECISION more conservative in their predicted probabilities, while the predicted probabilities by MetaTeacher are closer to 0 compared to them. From the visualization results, it can be seen that MetaTeacher is more accurate.

## C   More Discussions

### C.1   Training Time Comparison

The training runtime of MetaTeacher is compared with DECISION [1] and MME [8] on a single NVIDIA 3090Ti GPU over the transfer scenarios *NIH-CXR14, CheXpert, MIMIC-CXR* to *Open-i* and *NIH-CXR14, CheXpert, MIMIC-CXR* to *Google-Health-CXR*. The results are shown in Table 6. It can be observed that our method MetaTeacher is slightly slower than the approach of multi-source-free domain adaptation (*e.g.*, DECISION). Although multi-source-free domain adaptation methods do not need to update a student model, they involve other complex designs (*e.g.*, DECISION need to do k-means clustering, and CAiDA [3] has a searching process of Semantic-Nearest Confident Anchor). Instead, MetaTeacher only involves some simple matrix calculation. The second observation is that MetaTeacher is slightly faster than the approach of semi-supervised domain adaptation (MME). This is because semi-supervised domain adaptation needs to train a model for each source domain, whilst

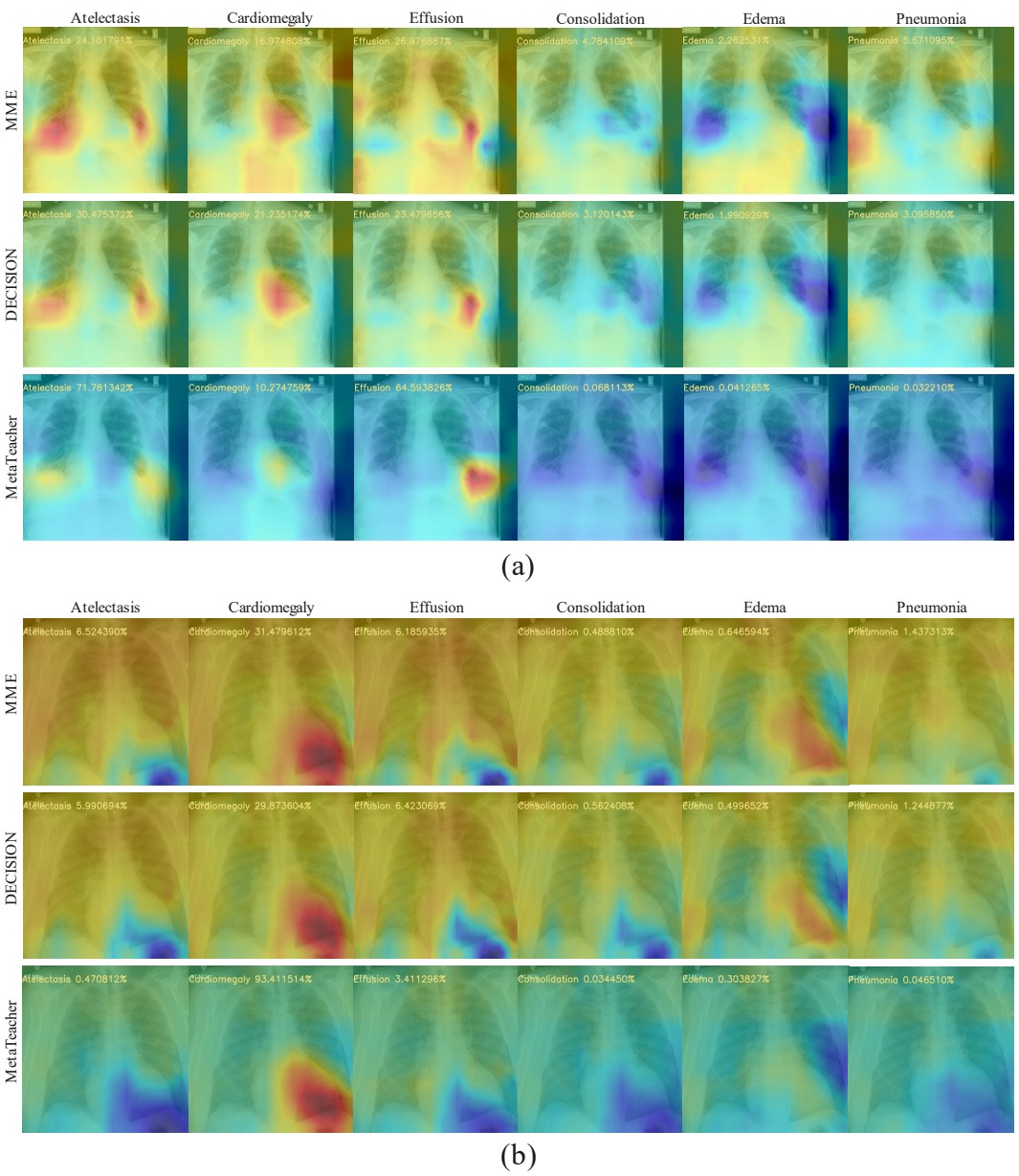

Figure 4: The Class Activation Map (CAM) [10] is used to perform visual ablation analysis on two chest x-ray images in *Open-i* dataset. The background color is blue, with red or yellow representing the disease location. The number on the top left corner of each image is the predicted probability for the corresponding disease. *Zoom in for best view.*

other complex computations are involved in their optimization process. Overall, our method has similar running speed as existing alternative methods.

## C.2 Probing the Behavior of Coordinating Weight

The coordinating weight is critical in our MetaTeacher framework. Firstly, for the upper-level optimization objective, it combines the predictions of multiple teachers to provide the updating direction for the student model. Secondly, for the lower-level optimization objective, we split coordinating weight into multiple vectors to provide different updating directions for each teacher. On the training of the transfer scenario *NIH-CXR14, CheXpert, MIMIC-CXR* to *Open-i*, we choose a

Table 6: Training time comparison. Metric: minutes.

| Method | DECISION [1] | MME [8] | MetaTeacher |
|---|---|---|---|
| NIH-CXR14, CheXpert, MIMIC-CXR to Open-i | 32 | 41 | 36 |
| NIH-CXR14, CheXpert, MIMIC-CXR to Google-Health-CXR | 33 | 43 | 38 |

Table 7: For a sample labeled as Atelectasis and Effusion classes, the weight changes before and after training.

| | Atelectasis | Cardiomegaly | Effusion | Consolidation | Edame | Pneumonia |
|---|---|---|---|---|---|---|
| Predictions for each teacher (pre-train) | 0.476732 | 0.267061 | 0.765137 | 0.190515 | 0.301033 | 0.168340 |
| | 0.243154 | 0.079328 | 0.481989 | 0.034837 | 0.400948 | 0.087144 |
| | 0.346073 | 0.345510 | 0.626205 | 0.021017 | 0.061996 | 0.042028 |
| Coordinating weight (pre-train) | **0.214039** | 0.149377 | 0.371503 | 0.733469 | 0.404418 | 0.445663 |
| | 0.022341 | 0.458379 | 0.377687 | 0.042648 | 0.404455 | 0.017554 |
| | 0.763619 | 0.392244 | 0.250810 | 0.223883 | 0.191127 | 0.536783 |
| Joint prediction (pre-train) | **0.371739** | 0.211779 | 0.623350 | 0.145928 | 0.295758 | 0.099113 |
| Predictions for each teacher (after-train) | **0.770673** | 0.327727 | 0.776080 | 0.161243 | 0.437272 | 0.377679 |
| | 0.430125 | 0.154535 | 0.124078 | 0.069540 | 0.047292 | 0.043537 |
| | 0.554779 | 0.255637 | 0.631193 | 0.052395 | 0.122772 | 0.199748 |
| Coordinating weight (after-train) | **0.950953** | 0.280765 | 0.984358 | 0.019614 | 0.000677 | 0.025918 |
| | 0.032633 | 0.704228 | 0.011930 | 0.196210 | 0.995239 | 0.943713 |
| | 0.016413 | 0.015007 | 0.003711 | 0.784177 | 0.004084 | 0.030369 |
| Joint prediction (pre-train) | **0.756016** | 0.204678 | 0.767763 | 0.057894 | 0.047864 | 0.056941 |

sample labeled with Atelectasis and Effusion disease classes to inspect the behavior of coordinating weight. As shown in Table 7, initially, each teacher, as well as their joint predictions ($\bar{y}_u^t$ in Eq. (12)), failed to predict the Atelectasis disease. During training, each teacher was updated, with teacher 1 acquiring the ability to predict Atelectasis disease (0.371739 to 0.756016). Meanwhile, the coordinating weight was also accordingly updated and assigned more weight to the Atelectasis class for teacher 1 (0.214039 to 0.950953). This process is summarized in Table 7.

### C.3 Performance Comparisons from Two Sources to Three Sources

Compared with multi-source-free domain adaptation methods, our MetaTeacher yields more significant gains in the multi-source transfer situation. On the two-teacher (Table 4) and three-teacher (Table 1) scenario, DECISION [1] method increases from 91.26% to 91.67% (a gain of 0.41%), CAiDA [3] method from 90.80% to 90.99% (0.19%), *vs.* our MetaTeacher from 91.98% to 92.84% (0.86%). This clearly shows that our performance gain is more significant than those by prior art methods.

For further validation, we have added a two-teacher transfer experiment: *CheXpert, MIMIC-CXR to Google-Health-CXR*. From the two-teacher case to the three-teacher transfer scenario (*NIH-CXR14, CheXpert, MIMIC-CXR to it Google-Health-CXR*), the performance gain of DECISION is 0.69%, vs. 1.35% by our MetaTeacher (see Table 8). This suggests that our method is superior at leveraging the diversity and complementary effect of multiple teacher models.

### C.4 Comparison with One-Teacher and One-Student Framework

Assuming no data privacy issue as discussed above, we experiment with a one-teacher one-student design where the component of adaptively training the teachers goes away naturally. We obtain the results of 89.97%/79.94%/75.38%/90.13%, inferior to 92.84%/82.69%/77.74%/91.98% by our MetaTeacher (corresponding to Tables 1,2,3,4 in the main paper). This is due to that each dataset presents unique characteristics in category imbalance and labeling error (*e.g.*, false negatives), resulting in different per-category qualities. Aggregating such datasets into one would introduce negative interference. Besides, using multiple teachers would reduce the learning difficulties of the entire classification problem in a spirit of divide-and-conquer principle, in addition to the opportunity of modeling the confidence per teacher. It should be noted that our multi-teacher setup is underpinned by the nature of our problem where data privacy protection is fundamentally critical (*i.e.*, data sharing

Table 8: Two-teacher and three-teacher transfer scenarios for the target domain *Google-Health-CXR*. Metric: AUROC.

| Method | DECISION [1] | MetaTeacher |
|---|---|---|
| CheXpert, MIMIC-CXR to Google-Health-CXR | 81.16 | 81.34 |
| NIH-CXR14, CheXpert, MIMIC-CXR to Google-Health-CXR | 81.85 | 82.69 |

across hospitals is typically banned). That being said, a specific teacher model would be trained using the training data of each individual hospital. This leads to the result of multiple teacher models in practice.

## C.5 Comparison with Semi-supervised Methods

Compared to the existing semi-supervised methods, our method requires less labeled data. For example, in the transfer scenario *CheXpert, MIMIC- CXR* to *NIH-CXR14*, for achieving 77.74% classification rate, the existing semi-supervised methods [2,4–6] require about 20,000 labeled samples (20% of the total), *vs.* our method only needing 500 labeled samples. Hence, our MetaTeacher is more data efficient and favored in practice.