# OpenReview forum: "MetaTeacher: Coordinating Multi-Model Domain Adaptation for Medical Image Classification"
_NeurIPS.cc/2022/Conference — NeurIPS 2022 Accept_

### Official Review · Reviewer_2Jhq · 2022-07-09

**Rating:** 6
**Confidence:** 4
**Soundness:** 2 fair
**Presentation:** 3 good
**Contribution:** 3 good

**Summary:**

This paper proposes a MetaTeacher for semi-supervised multi-source-free domain adaptation of medical image classification. The transfer learning process is modeled as a multi-teacher and one-student scheme. This model not only optimizes student, but also optimizes teachers through student’s feedback in the target domain. The optimization is based on meta-learning, which consists of two main part: coordinating weight learning, and bilevel optimization. Finally, experimental experiments on multi-label chest x-ray datasets empirically demonstrated the superiority of the proposed model against other SOTA methods.

**Questions:**

(1) Why do the authors design a multi-teacher one-student scheme? Whether we can set one-teacher and on-student framework? If yes, what is the performance when using one teacher?
(2) In Table 1, it can be seen that the proposed model still obtains promising performance when without using coordinating weight learning. Please analyze the reasons for this case.
(3) In Table, this experiment is only conducted on the proposed model, how does this is compared to other semi-supervised methods?



**Ethics Review Area:**

["I don’t know"]

**Limitations:**

See questions.



**Strengths And Weaknesses:**

Strengths:
(1) This paper studies a new problem setting, i.e., semi-supervised multi-source-free domain adaptation for multi-label medical image classification.
(2) A mutual feedback mechanism is designed based on meta-learning between the target model and the source models for more coherent learning and adaptation.
(3) The coordinating weight learning method is derived for dynamically revealing the performance differences of different source models over different classes.

Weaknesses:
(1) This is not clear why do the authors design a multi-teacher one-student scheme? Whether we can set one-teacher and on-student framework? If yes, what is the performance when using one teacher?
(2) In Table 1, it can be seen that the proposed model still obtains promising performance when without using coordinating weight learning.
(3) Table 5 shows the effects of the size of labeled target data on the transfer from NIH-CXR14, CheXpert, MIMIC-CXR to Open-i. However, this experiment is only conducted on the proposed model, how does this is compared to other semi-supervised methods?

---

> ### Author Response · Authors · 2022-08-01
> **Response to Reviewer 2Jhq**
>
> We thank the reviewer for positive comments and helpful feedback. We address the specific suggestions below.
>
> * * *
>
>
>
> **Q1: (1) Why do the authors design a multi-teacher one-student scheme? (2) Whether we can set one-teacher and one-student framework? If yes, what is the performance when using one teacher?**
>
> **A1:** This multi-teacher setup is underpinned by the nature of our problem where data privacy protection is fundamentally critical (I.e., data sharing across hospitals is typically banned). That being said, a specific teacher model would be trained using the training data of each individual hospital. This leads to the result of multiple teacher models in practice, rather than manually designed. We will further clarify.
>
> Assuming no above data privacy issue, as suggested we have experimented with a one-teacher one-student design (note this does not respect our problem setting as introduced in this work) where the component adaptive training the teachers goes away naturally. We obtained the results of **89.97%/79.94%/75.38%/90.13%**, inferior to **92.84%/82.69%/77.74%/91.98%** by ours (corresponding to Tables 1,2,3,4). This is due to that each dataset presents unique characteristics in category imbalance and labeling error (e.g., false negatives), resulting in different per-category qualities. Aggregating such datasets into one would introduce negative interference. Besides, using multiple teachers would reduce the learning difficulties of the entire classification problem in a spirit of divide-and-conquer principle, in addition to the opportunity of modeling the confidence per teacher.
>
> * * *
>
>
> **Q2: In Table 1, it can be seen that the proposed model still obtains promising performance when without using coordinating weight learning. Please analyze the reasons for this case.**
>
> **A2:** Great comments, thanks. As stated in Lines 208-213 of the paper, the coordinating weights \$W\$ plays two important roles: (1) Providing the directions for updating the student models, and (2) Providing the directions for updating each teacher. The variant **Ours (w/o mapping)** treats each teacher with equal weight, which can still bring performance gains over the source-only baseline due to the following reasons:
>
> (1) For student updating, averaging predictions from multiple teachers is beneficial for student performance, consistent with the finding by [1].
>
> (2) The fixed $W$ is also applied for teacher updating; In this case, the $W_{u}^{j}$ of Eq.(12) (Appendix) is a constant value. It is still applied during our derivation (Eq (13) to Eq(24)) (in Appendix) to benefit the optimization of the teacher models.
>
> We will further clarify.
>
> * * *
>
> **Q3: In Table, this experiment is only conducted on the proposed model, how does this is compared to other semi-supervised methods?**
>
> **A3:** Good suggestion, thanks. **Compared to existing semi-supervised methods, our method requires less labeled data.** For example, for the transfer scenario **CheXpert, MIMIC- CXR to NIH-CXR14**, to achieve 77.74% classification rate, the existing semi-supervised methods [2,3,4,5] requires about **20,000** labeled samples (20% of the total), vs. our method only needs **500** labeled samples. Hence, our method is more data efficient and preferred in practice.
>
> * * *
>
> **References**
>
> [1] You, Shan, et al. "Learning from multiple teacher networks." ACM SIGKDD2017.
>
> [2] Liu, Fengbei, et al. "Self-supervised mean teacher for semi-supervised chest x-ray classification." International Workshop on Machine Learning in Medical Imaging. 2021.
>
> [3] Liu, Fengbei, et al. "ACPL: Anti-Curriculum Pseudo-Labelling for Semi-Supervised Medical Image Classification." CVPR2022.
>
> [4] Liu, Quande, et al. "Semi-supervised medical image classification with relation-driven self-ensembling model." IEEE TMI (2020).
>
> [5] Aviles-Rivero, Angelica I., et al. "Graphxnet Chest X-Ray Classification Under Extreme Minimal Supervision." MICCAI(2019).
> * * *
>
> **Final remark:** We hope that our responses have resolved all the questions and concerns. In the
> revised version, we will add the above analysis for consolidating our work. Please feel free to let
> us know for more questions/comments/concerns if any.

---

### Official Review · Reviewer_6gX3 · 2022-07-10

**Rating:** 4
**Confidence:** 3
**Soundness:** 2 fair
**Presentation:** 2 fair
**Contribution:** 1 poor

**Summary:**

1. The authors proposed a new problem setting, i.e., semi-supervised multi-source-free domain adaptation (SMDA) for multi-label medical image classification.

2. A novel framework MetaTeacher based on a multi-teacher and one-student scheme is introduced to solve the proposed SMDA problem.

3. A coordinating weight learning method is derived for dynamically revealing the performance differences of different source models over different classes. It is integrated with the semi-supervised bilevel optimization algorithm for consistently updating the teacher and student models.

**Questions:**

Please refer to the weakness part.

**Ethics Review Area:**

["I don’t know"]

**Limitations:**

The authors have adequately addressed the limitations and potential negative societal impact of their work.

**Strengths And Weaknesses:**

Strength:

The proposed Semi-supervised Multi-source-free Domain Adaptation (SMDA) setting sounds interesting in the context of medical image classification.

Weakness:

My major concern lies in the effectiveness of the proposed solution for SMDA.

In Tables 2-4, it is obvious that the proposed approach (ours (all)) only performs slightly better than DECISION [1] on the transfer to Google-Health-CXR, NIH-CXR14, and Open-i. Considering DECISION is the state-of-the-art multi-source-free domain adaptation method, I think the proposed solution cannot reflect well on the value of SMDA because the multi-source-free domain adaptation approach can already well tackle the SMDA setting.

---

> ### Author Response · Authors · 2022-08-01
> **Response to Reviewer 6gX3**
>
> **Q1: I think the proposed solution cannot reflect well on the value of SMDA**
>
> **A1:** Thanks. Please note that our first important contribution is a novel practically critical problem setting, namely Semi-Supervised Multi-Source Domain-Free Adaptation (SMDA), for multi-label medical image classification. Further, our MetaTeacher method is superior in comparison to SOTA solutions:
>
> **(1) Significant performance gain in medical image classification:** As shown on the transfer scenarios in paper Tables 1-4, our method outnumbers the best competitor DECISION by 1.17%, 0.84%, 0.69%, and 0.72%, which we consider is significant. In comparison, we witness a gain of **0.8%** in supervised medical image classification over the three years from 2019 to 2021 as shown in the table below. Hence, the gain our method achieves should be considered as significant.
>
> Table1: AUROC comparison between supervised SOTA approaches trained with 100% of labelled data on NIH-CXR14.
> |Method|years|AUROC|
> |-|-|-|
> |Ma et al. [1]|2019|81.7|
> |Hermoza et al. [2]|2020|82.1|
> |S2MTS2 [3]|2021|82.5|
>
> **(2)More significant gain in multi-source transfer situation.** On the two-teacher (Table 4) and three-teacher (Table 1) scenario, DECISION increases from 91.26% to 91.67% (a gain of 0.41%), CAiDA from 90.80% to 90.99% (0.19%), vs. our MetaTeacher from 91.98% to 92.84% (0.86%). This clearly shows that our gain is more significant than those by prior SOTA methods. For further validation, we have added a two-teacher transfer experiment: **CheXpert, MIMIC-CXR to Google-Health-CXR**. From the two-teacher case to the three-teacher transfer scenario (NIH-CXR14, CheXpert, MIMIC-CXR to Google-Health-CXR), the performance gain of DECISION is 0.69%, vs. 1.35% by our MetaTeacher. This suggests that our method is superior at leveraging the diversity and complementary effect of multiple teacher models.
>
> Table 2: Two-teacher and three-teacher transfer scenarios for the target domain Google-Health-CXR
>
> |Method|DECISION|OURS|
> |-|-|-|
> |CheXpert, MIMIC-CXR to Google-Health-CXR|81.16|81.34|
> |NIH-CXR14, CheXpert, MIMIC-CXR to Google-Health-CXR|81.85|82.69|
>
> **(3) Stronger at locating the symptoms with practical significance.** From the analysis visualized in the **Appendix**, it is evident that the visualization background of our MetaTeacher is much clearer than DECISION, along with higher confidence in the predictions of samples clearly with and without disease. In practice, such high-confidence results are preferred by physicians.
>
> **(4) Higher fault tolerance.** Let's consider an extreme case where there is a single sample for a specific class, only one teacher model makes the correct prediction, while all the other models fail (This is possible in case of wrong source domain labels). In this case, the current multi-source-free domain adaptation methods would suffer with a tendence of making false predictions. In contrast, our MetaTeacher is likely to excel. As illustrated in Figure 3 (**Appendix**), our model can correctly predicte the disease Atelecsis even though none of the source domain teacher models succeed.
>
> **(5) Comparing with a strong combination of semi-supervised learning and multi-source domain adaptation (MSDA).** We have now additionally compared with an improved DECISION model (denoted as DECISION++): First fine-tuning all source domain teacher models using the target-domain labeled training data, and then using them to train a stronger DECISION model. Experimental results in the Table below show that our MetaTeacher can still surpass both variants of DECISION by a clear margin. This suggests that our method goes beyond combining existing semi-supervised learning and multi-source domain adaptation.
>
> Table 3 Comparison between MetaTeacher and DECISION under the same known conditions.
> |Method|DECISION|DECISION++|OURS|
> |-|-|-|-|
> |NIH-CXR14, CheXpert, MIMIC-CXR to Google-Health-CXR | 81.85|81.99|82.69|
> |CheXpert, MIMIC-CXR to NIH-CXR14| 77.05|77.28| 77.74|
> |NIH-CXR14, CheXpert to Open-i|91.26|91.50|91.98|
> ***
> **References**
>
> [1] Ma, Congbo, Hu Wang, and Steven CH Hoi. "Multi-label thoracic disease image classification with cross-attention networks." International conference on medical image computing and computer-assisted intervention. Springer, Cham, 2019.
>
> [2] Hermoza, Renato, et al. "Region proposals for saliency map refinement for weakly-supervised disease localisation and classification." International Conference on Medical Image Computing and Computer-Assisted Intervention. Springer, Cham, 2020.
>
> [3] Liu, Fengbei, et al. "Self-supervised mean teacher for semi-supervised chest x-ray classification." International Workshop on Machine Learning in Medical Imaging. Springer, Cham, 2021.
> ***
> **Final remark:** We hope that our responses have resolved all the questions and concerns. In the revised version, we will add the above analysis for consolidating our work. Please feel free to let us know for more questions/comments/concerns if any.

---

> > ### Comment · Reviewer_6gX3 · 2022-08-08
> > **Thank you for the clarifications**
> >
> > Personally, I'm satisfied with authors' responses, which are mainly about the improvements brought by the proposed method. However, I still cannot get the significance of the proposed new setting, i.e., semi-supervised multi-source-free domain adaptation (SMDA). In my opinion, the proposed approach ought to perform significantly better than previous baselines in SMDA, as those baselines were never specifically optimized for SMDA. However, the relative improvements are fewer than 1% in most cases, making me doubt the practical value of SMDA.

---

> > > ### Author Response · Authors · 2022-08-08
> > > **Response to Reviewer 6gX3**
> > >
> > > Really appreciate the reviewer for further feedback and interaction.
> > >
> > > First, we would like to stress that our proposed semi-supervised multi-source-free domain adaptation (SMDA) problem is novel yet significantly more challenging, fully respecting well the real-world situations (data sharing across hospitals is strictly restricted, along with sparse labelled data available in a target domain), as extensively discussed in our paper.
> > >
> > > Also, we consider that the performance gain by our method over the state-of-the-art alternatives (e.g., DECISION, DECISION++) is already non-trivial. Specifically, whilst the most related art model DECISION is not designed for the proposed SMDA problem, we have already adapted it accordingly to DECISION++ (First fine-tuning all source domain teacher models using the labeled target-domain training data, and further leveraging these tuned models to train a stronger variant). Indeed, the small gap between DECISION and DECISION++ in the range of [0.14%, 0.24%] exactly indicates the genuine challenge degree of this new problem, despite this has been the best possible existing solution further equipped with the strong fine-tuning baseline approach. In our community, fine-tuning is still a gold-standard baseline often yielding strong performance. In this context, our model’s gain of [0.69%, 0.84%] over DECISION and further gain of [0.46%, 0.70%] over DECISION++ are meaningful and non-trivial to achieve. We appreciate that the reviewer can take into account all the facts comprehensively in the final review evaluation.

---

### Official Review · Reviewer_5gnh · 2022-07-12

**Rating:** 6
**Confidence:** 4
**Soundness:** 3 good
**Presentation:** 3 good
**Contribution:** 3 good

**Summary:**

The paper presents a meta-learning approach for semi-supervised multi-domain source-free domain adaptation (SFDA). As first contribution, the authors extend the multi-domain SFDA problem to a semi-supervised learning scenario where a few labeled examples of the target domain are provided in addition to the unlabeled ones. As second contribution, they propose a meta-learning framework where a student learns from multiple teachers, each one trained on different source data. The student is trained to predict the correct class for labeled examples and, for unlabeled examples, to be consistent with a weighted average of the teachers' predictions where weights are learned. A bilevel optimization strategy is employed to update the parameters of the student and the teachers. The proposed method is evaluated on a multi-label chest X-ray classification task using five datasets. Results show that the method yield some improvements compared to recent approaches for SFDA.

**Questions:**

* Please provide some comparison in terms of computational complexity

* Give an example or explain what the behavior of the method for predicting W.



**Limitations:**

The limitations of the method are not mentioned in the paper.

**Strengths And Weaknesses:**

Strengths:

* Novelty: Although it borrows from previous works like Meta Pseudo Labels, the overall method proposed in the paper is novel. In particular, the coordinating weight learning is most original. The extension of multi-source SFDA problem to a semi-supervised setting is also interesting, even if it relates to other machine learning problems like domain generalization and continual learning.

* Experiments and results: The experimental evaluation of the method is comprehensive. It uses five large datasets and compares against several baselines, ablation variants and recent approaches. While the method's performance is only slightly better than competing approaches, it seems to provide better predictions when looking at the visualization examples.

Weaknesses:

* A potential weakness of the method is the high computational complexity brought by the bilevel optimization. It would be useful to compare the runtime of tested approaches, in addition to their accuracy.

* Figure 2 indicates that varying beta and gamma does not improve the performance much and that the corresponding loss terms might not be so useful.

* The coordinating weight learning is interesting, however experiments do not really study this component. If possible, it would be interesting to show what are weights for some examples of different sources.

*  I feel that the related works in the Supplemental materials are more relevant than those in the main paper, as they focus on SFDA and compared methods of the experiments. I recommend the authors to add them to the main paper (or swap the content if lacking space).

Minor comments:

* Introduction: define multi-label classification.

* Section 3.2: pesudo-label --> pseudo-label

* Eq (5): min_{theta_S} --> argmin_{theta_S}  ?

* Algorithm 1, line 5 (Sup.mat.): missing Theta_s on the right side ?

---

> ### Author Response · Authors · 2022-08-01
> **Response to Reviewer 5gnh**
>
> We thank reviewer for positive comments and appreciation for our work.
> ***
> **Q1: Please provide some comparison in terms of computational complexity.**
>
> **A1:** Thanks for the great comment. Following the suggestion, we have now compared the **training** runtime of MetaTeacher, DECISION and MME on a single NVIDIA 3090Ti GPU over the transfer scenarios **NIH-CXR14, CheXpert, MIMIC-CXR to Open-i** and **NIH-CXR14, CheXpert, MIMIC-CXR to Google-Health-CXR**. The results are shown in Table 1 below.
>
> **Table 1** Training running time comparison on two transfer scenarios.
>
> |Methods|NIH-CXR14, CheXpert, MIMIC-CXR to Open-i|NIH-CXR14, CheXpert, MIMIC-CXR to Google-Health-CXR|
> |---|-|-|
> |DECISION|32min|33min|
> |MME|41min|43min|
> |MetaTeacher(ours)|36min|38min|
>
> **(1) Slightly slower than multi-source-free domain adaptation (e.g.,DECISION).** Although multi-source-free domain adaptation methods do not need to update a student model, they involve other complex designs (e.g., DECISION need to do k-means clustering, and CAiDA has a searching process of Semantic-Nearest Confident Anchor). Instead, our method only involves a simple matrix calculation. **(2) Slightly faster than Semi-supervised domain adaptation (MME).** This is because semi-supervised domain adaptation needs to train a model for each source domain, whilst other complex computations are involved in their optimization process. Overall, our method has similar running speed as existing alternative methods.
> ***
>
> **Q2: Give an example or explain what the behavior of the method for predicting W.**
>
> **A2:** Thanks. The weight \$W\$ is critical in the MetaTeacher framework. **Firstly**, for the upper-level optimization objective, it combines the predictions of multiple teachers to provide the updating direction for the student model (Lines 195-198). **Secondly**, for the lower-level optimization objective, we split \$W\$ into multiple vectors to provide different updating directions for each teacher (Lines 170-173). In our experiments, on the training of the transfer scenario **NIH-CXR14, CheXpert, MIMIC-CXR to Open-i**, we have noted \$W\$ w.r.t. **a sample labeled with Atelectasis and Effusion disease classes** (Table 2 below). Initially, each teacher, as well as their joint predictions ($\bar{y}_{u}^{t}$ in Eq(12)(Appendix)), failed to predict the Atelectasis disease. During training, each teacher was updated, with teacher1 gained the ability to predict Atelectasis disease (0.371739-\>0.756016); Meanwhile, \$W\$ was also accordingly updated and assigned more weight to the Atelectasis class for teacher1 (0.214039-\>0.950953). This process is summarized in the table below. We will add this analysis to the revised version.
>
> **Table 2** For a sample labeled as Atelectasis and Effusion classes, the weight changes before and after training.
> ||Atelectasis|Cardiomegaly|Effusion|Consolidation|Edame|Pneumonia|
> |-|-|-|-|-|-|-|
> |**Predictions for each teacher (pre-train)**|0.476732|0.267061|0.765137|0.190515|0.301033|0.168340|
> ||0.243154|0.079328|0.481982|0.034837|0.400948|0.087144|
> ||0.346073|0.345510|0.626205|0.021017|0.061996|0.042028|
> |**W (pre-train)**|**0.214039**|0.149377|0.371503|0.733469|0.404418|0.445663|
> ||0.022341|0.458379|0.377687|0.042648|0.404455|0.017554|
> ||0.763619|0.392244|0.250810|0.223883|0.191127|0.536783|
> |**Joint predictions (pre-train)**|0.371739|0.211779|0.62335|0.145928|0.295758|0.099113|
> |**Predictions for each teacher (after-train)**|0.770673|0.327727|0.776080|0.161243|0.437272|0.377679|
> ||0.430125|0.154535|0.124078|0.069540|0.047292|0.043537|
> ||0.554779|0.255637|0.631193|0.052395|0.122772|0.199748|
> |**W (after-train)**|**0.950953**|0.280765|0.984358|0.019614|0.000677|0.025918|
> ||0.032633|0.704228|0.011930|0.196210|0.995239|0.943713|
> ||0.016413|0.015007|0.003711|0.784177|0.004084|0.030369|
> |**Joint prediction (after-train)**|**0.756016**|0.204678|0.767763|0.057894|0.047864|0.056941|
> ***
>
> **Other comments:**
>
> **Q1: Figure 2 indicates that varying beta and gamma does not improve the performance much and that the corresponding loss terms might not be so useful.**
>
> **A1:** Due to space limitations, we did not give the specific values in Figure 2. On the transfer scenario **NIH-CXR14, CheXpert, MIMIC-CXR to Open-i**, it is shown that without adding β and γ, the results were 92.49% and 92.29%, which is 0.35% and 0.55% lower than the final result 92.84% (Table 1 in paper). This suggests their good effectiveness. Please also see our response to Q1 of Reviewer 6gX3.
>
> **Q2: Paper structure suggestions and typos.**
>
> **A2:** Many thanks. We will refine and correct them in the revised version.
> ***
> **Final remark:** We hope that our responses have resolved all the questions and concerns. In the revised version, we will add the above analysis for consolidating our work. Please feel free to let us know for more questions/comments/concerns if any.

---

> > ### Comment · Reviewer_5gnh · 2022-08-08
> > **Thank you for the clarifications**
> >
> > I appreciate the authors' detailed answers which have addressed my concerns about computational complexity and shed some light on the behaviour of coordination mechanism.

---

> > > ### Author Response · Authors · 2022-08-08
> > > **Thank you for your comments.**
> > >
> > > Thank you for your kind answer. We are glad that our responses could clarify all your concerns. In case there are no further questions, we would appreciate if you could improve your score.

---

### Author Response · Authors · 2022-08-01
**General Response**

We thank all the reviewers for insightful comments on our work. We appreciate the following positive feedback:

1. The multi-source SFDA problem setting is interesting (Reviewer 5gnh, 6gX3). The paper studies a new problem setting. (Reviewer 2Jhq).
2. The proposed method is novel and interesting (Reviewer 5gnh, 6gX3).
3. The coordinating weight learning is novel (Reviewer 5gnh). A mutual feedback mechanism is designed (Reviewer 2Jhq).
4. The experimental evaluation is comprehensive (reviewer 5gnh).

Given the wide diversity in data collection across hospitals, we propose a practical yet under-studied new problem for multi-label medical image classification, namely **Semi-Supervised Multi-Source Domain-Free Adaptation (SMDA).** We further introduce a novel approach named MetaTeacher in a multi-teacher and one-student framework. For model training, we leverage a bilevel optimization strategy for alternatively updating the student and teacher models, characterized by a coordinating weight scheme for adaptively calibrating the learning directions of both student and teacher models.

We address all the comments of each reviewer in detail below. We thank all reviewers for the first round of comments, please feel free to let us know if further clarifications and experiments are needed. We would really appreciate it if the reviewers could consider raising the scores accordingly in reflection of our responses and updates.

---

### Meta-Review · Area_Chair_pPuV · 2022-08-28

**Recommendation:** Accept
**Confidence:** Certain

**Metareview:**

The paper proposes a model for a multiple teacher, single student setting for medical image classification.

The reviewers where split, with two reviewers leaning towards accept and one leaning towards reject. The main criticism of the negative reviewers is that the proposed model only slightly outperforms the state of the art. Given the extensive experimental evaluation and the fact that the proposed method consistently outperforms the state of the art, the improvement should be statistically significant. The negative reviewer has acknowledged the improvement in the discussion phase. A second criticism was the lack of significance of the proposed learning setting. As the reviewers find this setting novel and due to the relevance in the medical domain, I vote to accept the paper.

**Award:**

No

---

### Decision · Program_Chairs · 2022-09-14

Accept